# Radiation Exposure from GEP NET Surveillance

**DOI:** 10.3390/cancers16020427

**Published:** 2024-01-19

**Authors:** Jordan Iannuzzi, Caitlin T. Yeo, Vicky Parkins, Dean Ruether, Errol Stewart, Denise Chan, Janice Pasieka, Kirstie Lithgow

**Affiliations:** 1Department of Medicine, Cumming School of Medicine, University of Calgary, Calgary, AB T2N 4N1, Canada; 2Department of Surgery, Cumming School of Medicine, University of Calgary, Calgary, AB T2N 4N1, Canada; 3Alberta Health Services, Foothills Medical Centre, Calgary, AB T2N 2T9, Canada; 4Department of Radiology, Cumming School of Medicine, University of Calgary, Calgary, AB T2N 4N1, Canada

**Keywords:** neuroendocrine tumor, diagnostic imaging, surveillance

## Abstract

**Simple Summary:**

Neuroendocrine tumors of the bowel and pancreas are slow-growing cancers that, if identified early, can be treated with surgery. However, even if surgery is successful, many patients must undergo surveillance imaging to check for recurrence; this involves CT or MRI scans every one to two years for up to ten years. CT scans are associated with radiation exposure, which can increase the risk of additional cancers later in life. We sought to assess real life radiation exposure associated with surveillance imaging for patients at our center with surgically removed neuroendocrine tumors. We reviewed patient charts to collect information about the scans they underwent and calculated the total radiation dose they received during follow-up. We found that the average total radiation dose was in the range associated with increased cancer risk. We suggest strategies to minimize radiation risk should be considered in future recommendations for surveillance of neuroendocrine tumors.

**Abstract:**

Background: Neuroendocrine tumors (NET) are neoplasms that secrete peptides and neuroamines. For gastroenteropancreatic (GEP) NET, surgical resection represents the only curative option. Ten-year imaging surveillance programs are recommended due to long time-to-recurrence following resection. We performed retrospective chart review evaluating radiation exposure and practice patterns from surveillance of completely resected GEP NET. Methods: We performed a retrospective cohort study of cases with well-differentiated GEP NET from January 2005 to July 2020. Location of primary, modality of imaging, and duration of follow-up were collected. Dosimetry data was collected to calculate effective dose. Results: 62 cases were included with 422 surveillance scans performed. Cross-sectional imaging was used in 82% and functional imaging was used in 18% of scans. Mean number of scans per year was 1.25 (0.42–3). Mean total effective dose was 56.05 mSv (SD 45.56; 0 to 198 mSv) while mean total effective dose per year was 10.62 mSv (SD 9.35; 0 to 45 mSv). Over the recommended ten years of surveillance the estimated total effective dose was 106 mSv. Conclusions: Surveillance of completely resected GEP NET results in cumulative radiation doses in the range associated with secondary malignancy development. Strategies to minimize radiation exposure in surveillance should be considered in future guideline development.

## 1. Introduction

Neuroendocrine tumors (NET) are a heterogenous group of malignant neoplasms that secrete peptides and neuroamines [1]. For patients with gastroenteropancreatic (GEP) NET, surgical resection represents the only curative option [2,3]. Following surgical resection, GEP NET typically have long periods of disease-free survival and time-to-recurrence [2,4,5]; median time to recurrence has been reported as seven to eight years [2,6]. Therefore, even for patients who have undergone complete surgical resection of GEP NET, long-term surveillance schedules including cross-sectional imaging with or without biochemistry are recommended [2,3]. However, there is no consensus regarding optimal follow-up in this patient population [7]. The most current guidelines recommend cross-sectional imaging with either triple phase CT or MRI every one to two years for at least ten years for most patients [2,8,9,10,11] (Appendix A). However, these recommendations raise theoretical concerns regarding cumulative radiation exposure over time, particularly if contrast-enhanced CT scans are performed. A linear and causal relationship between ionizing radiation exposure and human cancer risk has been established [12], and doses of radiation exceeding 50 mSv are associated with increased risk of secondary malignancy [13,14]. Given a CT scan of the abdomen and pelvis can give an effective dose of 4 to 25 mSv [15,16,17], long-term imaging surveillance may be associated with risk of radiation-induced malignancy. This is an important consideration when deciding on an optimal imaging surveillance strategy in our patient population, particularly in younger individuals [13,15].

Previous authors have raised theoretical concerns about the radiation associated with surveillance imaging for completely resected GEP NET [2], and cross-sectional imaging with CT scan has been cited as the preferred modality of surveillance imaging [7]. However, the cumulative radiation exposure associated with real world follow-up surveillance in this patient population has not been previously assessed. Our primary objective was to measure the cumulative radiation exposure associated with real-world post-operative surveillance imaging at our tertiary center. Our secondary aim was to characterize local practice patterns of imaging surveillance in relation to current guidelines. 

## 2. Materials and Methods

We retrospectively reviewed clinical and imaging records from patients diagnosed with well-differentiated GEP NET at our centre (Tom Baker Cancer Centre, University of Calgary) from January 2005 to July 2020 inclusive. The study was approved by our institutional research ethics board (HREBA.CC-20-0090). Inclusion criteria were GEP NET that had been fully resected (defined as surgical removal with curative intent), stages I to III disease at diagnosis, age 18 or older, and follow-up care and surveillance in our neuroendocrine cancer clinic. The post-resection surveillance protocol is in keeping with the Commonwealth Neuroendocrine Tumor Research Collaborative (CommNETs) guideline recommendations [2]. Exclusion criteria included incompletely resected NET or stage IV disease at presentation, G1 appendiceal NET less than 1 cm, R0G1T1 rectal NET, G1 stage 1 midgut NET, insulinoma, neuroendocrine carcinoma, or concurrent diagnosis of non-NET metastatic cancer (Table 1). Potentially eligible cases were identified by a data analyst from the Alberta Cancer Registry. Demographic information retrieved included date of birth, sex, date of surgical resection, primary tumor site, and date of last known follow-up. Our electronic medical record does not collect data on race and therefore we were unable to retrieve this data. Information was retrieved from patient electronic medical records. Prior to further data extraction, cases were screened for eligibility and excluded as appropriate. For each eligible case, we collected data from our imaging database (IMPAX) for all follow-up scans that were performed for the purposes of post-operative surveillance. Imaging performed for other indications was excluded. For each surveillance scan we collected the date and modality of imaging. When multiple imaging modalities were performed concurrently for the purposes of the same follow-up appointment (i.e., contrast-enhanced CT with octreotide scan) this was considered a single surveillance scan and the date of the earliest scan was recorded. 

For all contrast-enhanced CT scans, dose length product (DLP) and tube voltage (kV) were retrieved from the dose report. This information was used to calculate effective dose in millisieverts (mSv) for each scan by multiplying the DLP by the appropriate conversion factor as described by Deak et al. [18]. For contrast CT scans which included both the abdomen and pelvis, the conversion factor for abdomen was used. 

Octreotide 111 In-Pentreotide scans involved 100 MBq IV of 111 In-Pentetreotide with 24 h delayed whole-body images and SPECT/CT of the abdomen and pelvis. SPECT/CT imaging involves a low dose localizer CT for attenuation correction and localization. I-123 MIBG (metaiodobenzylguanidine) imaging involved injection of 370 MBq 123 I- MIBG intravenously. Delayed whole-body planar imaging with SPECT/CT of the abdomen and pelvis with low dose CT was used for attenuation correction and localization. For both Octreotide 111 In-Pentreotide and I-123 MIBG scans, additional SPECT/CT of other sites may have been performed if clinically indicated. The effective doses for Octreotide 111 In-Pentreotide and I-123 MIBG imaging were estimated at 5.4 and 4.8 mSV respectively based on the international committee for radiological protection (ICRP) [19,20]. 

18F-FDG PET/CT scans involved injection of 400 MBq 18F-FDG intravenous infusion including imaging with low dose CT for attenuation correction and localization from the skull base to the mid thighs. The effective doses for 18F-FDG PET/CT were calculated using previously described methodology [21].

68Ga-DOTATATE PET/CT scans involved 200 MBq intravenous injection with low dose CT for attenuation correction and localization from the top of the head to the mid thighs. The effective doses for 68Ga-DOTATATE PET/CT were calculated using methodology described by Walker et al. [22]. The effective dose from the associated low dose CT scan was included in the total reported dose from this modality. 

## 3. Results

The initial case retrieval yielded 387 cases. 62 cases met criteria for data extraction and analysis. Reasons for exclusions were: not followed up at our centre (n = 268); metastatic disease at presentation (n = 44); and low risk subgroup (n = 13) including G1 appendiceal NET < 1 cm and R0G1T1 rectal NET. The mean age at diagnosis was 58 years (SD 14 years, range 21 to 82 years) and 55% (n = 34) cases were female. Location of primary was small bowel in 58% (n = 36), appendix in 23% (n = 14), pancreas in 13% (n = 8), large bowel in 3% (n = 2), and rectum in 3% (n = 2). Mean length of follow-up was 71 months (SD 33 months) (Table 2). Among the eight cases that had a resection date 10 or more years ago, the mean length of follow-up was 10.4 years (range 3.5 to 12.3 years, SD = 2.9 years).

There were 422 surveillance scans performed in the cohort. Imaging modalities included cross-sectional imaging (MRI and contrast enhanced CT) and nuclear medicine imaging. Cross-sectional imaging was most commonly performed with CT and MRI representing 41% (n = 173) and 41% (n = 172) of all surveillance scans respectively. Nuclear medicine imaging was used in 18% (n = 77) of scans; Octreotide 111 In-Pentreotide (n = 51), I-123 MIBG (n = 14), 68Ga-DOTATATE PET (n = 9), and 18F-FDG PET (n = 3) (Figure 1). 

For the functional imaging, 60% (n = 37) of the 62 patient cases had at least one nuclear medicine imaging study performed during the follow-up interval. In those 37 cases, 29% (n = 18) had nuclear medicine imaging at multiple time points, and 31% (n = 19) had nuclear medicine imaging only once during the follow-up period. The remaining 40% did not undergo nuclear medicine imaging. The first post-operative surveillance was the most common timeframe for performing functional imaging (n = 23). In instances where nuclear imaging was not accompanied by recent cross-sectional imaging, a low dose SPECT CT was performed.

The mean number of surveillance scans was 7 (SD 3, range 2 to 14) and the mean number of surveillance scans per year of follow-up was 1.25 (SD 0.51, range 0.42 to 3.00). The mean effective dose per contrast-enhanced CT scan was 16.90 mSv (SD 12.17, range 3.39 to 57.75 mSV). The average effective dose from 68Ga-DOTATATE PET, including the CT portion of the scan, was 13.85 mSv (SD 4.34 mSV, range 10.22 to 22.25). The mean total effective dose from all imaging was 56.05 mSV (SD 45.56 mSV, range 0 to 198.22 mSV). The mean total effective dose per year of follow-up was 10.62 mSV (SD 9.35 mSV, range 0 to 44.05 mSV). In linear regression analysis, age at diagnosis was not found to be a significant predictor of effective dose per year (F(1,60) = 2.069, *p* = 0.156) with R^2^ of 0.033. The location of primary was also not a significant predictor of effective dose per year (F(4,57) = 1.492, *p* = 0.217), with R^2^ of 0.095

Using mean effective doses for contrast-enhanced CT and 68Ga-DOTATATE PET, we estimated cumulative radiation doses associated with the different guideline derived imaging protocols (Table 3). As all guidelines stated CT or MRI could be used interchangeably, and we have shown cumulative doses for each protocol using CT alone or CT and MRI in combination. Details of our calculations are shown in Appendix A.

## 4. Discussion

This is the first study evaluating real-world imaging surveillance of completely resected GEP NET and associated radiation exposure to patients. GEP NET are unique in the oncology population in that the majority of patients are surviving at 15 years [2]. Given the long survival seen in these patients, the potential harms associated with cumulative radiation exposure deserve our consideration. 

There is a linear and causal relationship between ionizing radiation exposure and human cancer risk [12,13,23,24]. Though whether this relationship exists at lower doses (less than 20–50 mSv) of radiation remains contentious, the risk of malignancy with doses exceeding 50 mSv is well established [12,14]. Without accounting for duration of follow-up, the average effective dose of radiation in our study exceeded 50 mSv. Extrapolation from the mean effective dose per year (10.62 mSv) yields an estimated effective dose of 106.20 mSV over a ten-year follow-up period, which was within the range of estimated effective doses for the imaging protocols listed in Table 3. 

There was variation in the radiation exposure from different guideline-derived recommendations and cross-sectional imaging protocols. This represents an area of possible improvement; further nuancing of recommendations based on expected timing of recurrence may reduce radiation exposure. For example, Strosberg el al. [25] reported a peak recurrence of pancreatic NET at two years, and no documented recurrence after seven years. In contrast, Cives et al. [5] (2018) showed consistent disease recurrence rates among stage 2 and 3 midgut NET for the first eight years of surveillance with a sharp drop off thereafter. There is also opportunity to tailor frequency of surveillance on the basis of tumor biology and disease stage. Previous authors have shown that tumor-specific prognostic factors such as histological grade, Ki-67 index/mitotic count, stage/size, nodal status, and site of origin influences the frequency of surveillance arranged by providers [7]. We suggest that dedicated review of NET cases at multidisciplinary rounds (such as those which occur at our center and other centers specializing in care of GEP NETs) can be useful for comprehensive assessment and accurate staging of NET cases in order to appropriately tailor surveillance schedules. Furthermore, we suggest preferential use of MRI over CT in some circumstances (i.e., surveillance of pancreatic NET in patients less than 50 years old at diagnosis) where resources permit, or alternating CT with MRI to increase the latency between ionizing radiation exposures as this also may impact secondary malignancy risk [26]. Finally, we emphasize that surveillance should not be performed for subgroups with low recurrence risk including N0 insulinomas, grade 1 stage I incidental midgut NET, and grade 1 appendiceal NET < 1 cm as outlined in the CommNETs guidelines [2].

Most cases in our study had small bowel primaries, in keeping with known epidemiology of GEP NET [27]. Cross-sectional imaging was the most used form of imaging surveillance with equal distribution between CT and MRI (Figure 1). The choice of cross-sectional imaging modality may be influenced by disease and patient factors. Triple phase contrast-enhanced CT is the preferred modality for the detection of intra-abdominal disease, whereas MRI is superior for the detection of hepatic metastases and imaging the pancreas [28]. This point was emphasized to our NET clinicians during a provincial tumor group meeting in November 2019 which may have impacted local practice patterns. However, we did not find that site of primary tumor was a significant predictor of effective dose per year. 

Nuclear medicine studies were utilized in 18% (n = 77) of scans and the first post-operative surveillance was the most common timeframe for performing functional imaging. Octreotide 111 In-Pentreotide scans were the most common functional imaging modality. Somatostatin receptor scintigraphy (SRS) at our institution may be performed with either Octreotide 111 In-Pentreotide or 68Ga-DOTATATE PET. 68Ga-DOTATATE PET has been available at our institution since 2020. Its use has been limited by resource constraints and therefore has not yet completely replaced Octreotide 111 In-Pentreotide scans, though we acknowledge that the use of Octreotide 111 In-Pentreotide scans is outdated and 68Ga-DOTATATE PET is preferred when evaluating NET. The average effective dose from 68Ga-DOTATATE PET in our study of 13.85 mSV was higher than from Octreotide 111 In-Pentreotide; more widespread use of 68Ga-DOTATATE PET may further contribute to longitudinal radiation exposure. The utility of SRS in detecting post-operative recurrence requires further evaluation, especially given the additional burden of radiation exposure associated with these modalities. 

The radiation exposure from long term surveillance of GEP NET is associated with an increased risk of secondary malignancy, especially in younger individuals [13,15,29,30]. However, in our study patient age at diagnosis was not a significant predictor of the mean effective dose per year. Our center follows the Comments [2] recommendations and our results indicate that our practices are generally in-line with these (and other contemporary guidelines) with respect to imaging frequency and duration of follow-up [8,9,10,11,28]. However, the average number of scans per person-year (1.1) suggests we may be over-surveilling some patients, representing an area of possible improvement. 

There is no consensus regarding the optimal timing and frequency of follow-up for GEP NET. This may be due in part to heterogeneous clinical behavior and paucity of data regarding recurrence rates [7]. Nonetheless, variation in recommendations from different society guidelines has been highlighted previously [7], and may contribute to confusion and frustration for health care providers. A previous interntational survey of NET health care providers from Australia, Canada, New Zealand, and the United States (published prior to the 2018 CommNETs guidelines [2]) demonstrated that a minority of respondents (17 to 38%) were “very familiar” with ENETS, ESMO, and NCCN society guidelines, and only 10 to 27% felt these guidelines were “very useful.” This survey demonstrated heterogeneity with respect to follow-up frequency and duration reported by providers. Follow-up every 6 months for the first two years and annually thereafter was most commonly chosen, and most respondents stated that they would opt for CT imaging as surveillance modality in post-operative years 0 to 5 [7]. The results of this study further highlight concerns around radiation exposure from imaging surveillance, as well as the potential benefit of harmonizating different society guidelines. 

Our study demonstrated heterogeniety in use of functional imaging. Results from the international survey detailed above [7] simlarily showed that NET providers utilize functional imaging in a minority of patients and the timeframe for performing this imaging is variable. These results likely reflect lack of formal recommendations by Comments [2] and varying recommendations by the other consensus guidelines with respect to functional imaging [8,9,10,11,28]. Clarity around use of functional imaging in future guidelines would be helpful for NET care providers. 

### Limitations

This is a single center experience and therefore reflects local practice patterns and resource availability that may not be generalizable to other centers. However, given that this audit showed our center’s practices are largely in-line with the most recent GEP NET surveillance guidelines, it is reasonble to assume that surveillance programs from other tertiary centers yield similar radiation exposure (especially given that CT was the preferred imaging modality for surveillance from the international survey highlighted above [7]). It is also important to note that this study looks at surveillance imaging performed over 18 years. The radiation dose associated with CT scans and injected activity of radiopharmaceuticals has changed over that time and will continue to evolve. Ongoing developments in technology and improved accessibility of MRI will hopefully lead to future reductions in radiation exposure.

We acknowledge that in addition to the total amount of radiation, the latency period between ionizing radiation exposures also contributes to secondary malignancy risk ([26]) and this was not accounted for in our methodology. 

The potential harms of radiation exposure from surveillance imaging must be weighed against the potential benefits of timely detection of recurrence, especially since poor tumor-specific prognostic factors are associated with increased surveillance [7]. As such, recurrence rates in our study population are a relevant metric. However, unfortunately, due to the retrospective nature of our study, we were not able to obtain sufficient clinical data to completely and accurately document recurrence rates in our study cohort. Furthermore, we did not have sufficient follow-up data to study rates of secondary malignancies. The real-world rates of secondary malignancies in this patient population are an important topic of future investigation and may be best answered through a larger epidemiological study. 

Each organ has different thresholds for radiation-induced cancer development [12]. The effective dose coefficients used in our study were roughly for the “abdominal organs”, not acknowledging the individualized surrounding organ dose. The use of individual tissue weighting factors and determining the of amount of organ-specific radiation of each scan for each patient would have been a complex undertaking and likely would have given limited additional information. This organ-specific information would be useful for assessment of interval cancer development in patients with fully resected GEP NET undergoing surveillance. However, this question was felt to be out of the scope of our study objectives and would be served best by an independent longitudinal study with control groups. 

## 5. Conclusions

This is the first study looking at cumulative radiation exposure associated with follow-up surveillance of patients with completely resected GEP NET. Our center’s practices are largely in-line with current guidelines, however, adherence to these guidelines can result in cumulative radiation doses associated with secondary cancer development. Providers should consider individual patient and disease factors when determining modality and frequency of surveillance imaging. Strategies to minimize radiation exposure during longitudinal surveillance of GEP NET should be considered by guideline authors and multidisciplinary teams. 

## Figures and Tables

**Figure 1 cancers-16-00427-f001:**
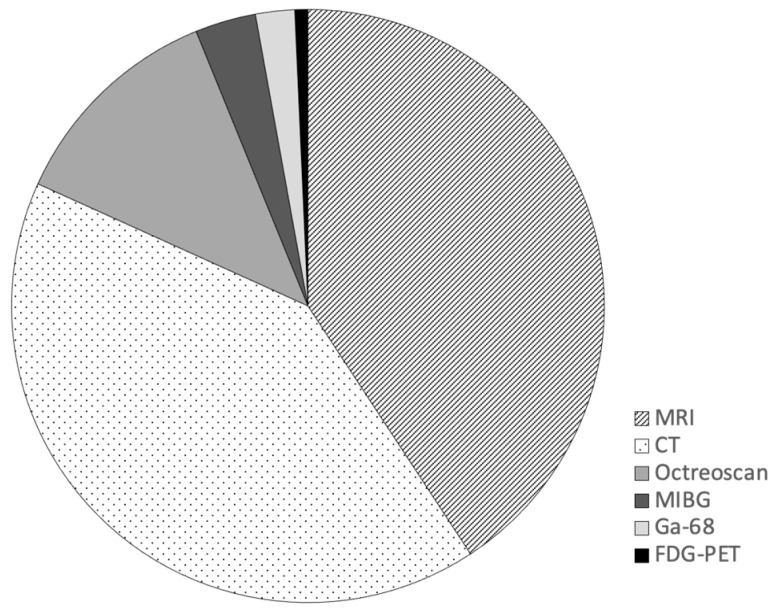
Imaging modalities used in post-operative surveillance of completely resected GEP-NET.

**Table 1 cancers-16-00427-t001:** Study inclusion and exclusion criteria.

Inclusion Criteria	Exclusion Criteria
Age 18 or older Any grade GEP NET Fully resected with curative intent Stage I II or III disease Follow-up care at our specialty NET clinic	Incompletely resected NET Stage IV disease at presentation G1 appendiceal NET less than 1cm R0G1T1 rectal NET Stage 1 grade 1 incidental midgut NET Insulinoma Neuroendocrine carcinoma Concurrent diagnosis of non-NET metastatic cancer

**Table 2 cancers-16-00427-t002:** Baseline characteristics of GEP NET cases.

Characteristics	Cases (n = 62)
Mean age (standard deviation; range)	58 years (14; 21−2)
Sex	F: 55% (n=34)
Location of primary	Small bowel 58% (n = 36) Appendix 23% (n = 14) Pancreas 13% (n = 8) Colon 3% (n = 2) Rectum 3% (n = 2)
Mean length of follow up in months (SD; range)	71 (33; 8−147)
Mean number of surveillance scans per case (range)	7 (2−14)
Mean number of scans/year of follow-up (range)	1.3 (0.4−3.0)
Surveillance scans/person year	1.1

**Table 3 cancers-16-00427-t003:** Estimated cumulative radiation exposure associated with imaging protocols derived from different consensus guidelines.

Guideline	Cross Sectional Imaging Protocol	Approximate Cumulative Effective Dose *
NANETS	CT only Alternating CT and MRI	190 mSV 90 mSV
ENETS	CT only Alternating CT and MRI	270 mSV 170 mSV
NCCN	CT only Alternating CT and MRI	120 mSV 60 mSV
ESMO	CT only Alternating CT and MRI	370 mSV 200 mSV
CommNETs	CT only Alternating CT and MRI	120 mSV 60 mSV

* Rounded to the nearest whole number in increments of 5.

## Data Availability

The data presented in this study are available upon reasonable request from the corresponding author.

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
