# Peer review of "Radiation Exposure from GEP NET Surveillance"

_cancers, 2024, doi:10.3390/cancers16020427_

Round 1

Reviewer 1 Report

Comments and Suggestions for Authors

This is the first study looking at cumulative radiation exposure associated with follow-up surveillance of patients with completely resected GEP NETs.  The authors found that adherence to guidelines for resected GEP NET can result in cumulative radiation doses associated with secondary cancer development. The authors conclude that strategies to minimize radiation exposure during longitudinal surveillance of GEP NETs should be considered by guideline authors and multi-disciplinary teams.  The authors also propose some strategies to minimize radiation exposure including the use of MRI scans which do not subjects patients to ionization radiation.

The manuscript is well written and should be of interest to readers of Cancers

I have several questions for the authors:

1)    What was the rate of cancer recurrence that was detected at the authors center using the surveillance?

2)    Do the authors know if any of the patients that were surveilled developed secondary malignancies and, if so, what type of secondary malignancies?

Author Response

I have several questions for the authors:

1)    What was the rate of cancer recurrence that was detected at the authors center using the surveillance?

2)    Do the authors know if any of the patients that were surveilled developed secondary malignancies and, if so, what type of secondary malignancies?

Thank you for your review. We agree these are important questions. Unfortunately, the retrospective nature of our study did not allow us to report recurrence rates in a way that would be accurate and complete. Furthermore, due to the latency between radiation exposure and secondary malignancy occurrence, we did not have sufficient follow-up data to report on this. We have expanded our limitations section to highlight these important issues.

Reviewer 2 Report

Comments and Suggestions for Authors

Neuroendocrine tumors are on the increase compared to a few years ago, when they were classified as carcinoids. They are ubiquitous diseases, which originate from argentaffin cells. They are quite common throughout the gastrointestinal system. Often in the small intestine and especially in the ileum they are multifocal. We absolutely agree on the fact that if discovered occasionally they are not metastatic and for this reason survival is long before secondaries appear. Fortunately, the age of onset is mostly around 60 but they can be observed at a younger age. Since surgical therapy allows eradication, recurrence, unlike adenocarcinomas, occurs after many years. The follow-up of these patients is obviously long so our colleagues have rightly asked themselves the question of how nociceptive ionizing radiation is for CT or for the infusion of radioactive drugs, performed for follow-ups and to what extent these treatments can , develop other types of neoplasms. In fact, several articles on this topic have been published in the literature, cited by colleagues, and the conclusion we can draw from them is that a greater risk of developing a secondary neoplasm due to exposure to ionizing radiation for tests conducted every three months is true. It is not certain that this is the case for tests carried out over longer periods. The researchers stuck to the guidelines, however they did not mention the dosage of chromogrania A, enolase, which are nevertheless useful. MRI can be useful for limiting radiation exposure, as also mentioned in the paper, and therefore certainly needs to be developed. the conclusions are consistent with the subject of discussion as well as the limitations they have extensively discussed. The bibliography is consistent and correct, the English can be made more fluent, tables and figures are consistent.

DOI: 10.1017/S1049023X1900520X

Comments on the Quality of English Language

the English needs to be a little more linear

Author Response

Neuroendocrine tumors are on the increase compared to a few years ago, when they were classified as carcinoids. They are ubiquitous diseases, which originate from argentaffin cells. They are quite common throughout the gastrointestinal system. Often in the small intestine and especially in the ileum they are multifocal. We absolutely agree on the fact that if discovered occasionally they are not metastatic and for this reason survival is long before secondaries appear. Fortunately, the age of onset is mostly around 60 but they can be observed at a younger age. Since surgical therapy allows eradication, recurrence, unlike adenocarcinomas, occurs after many years. The follow-up of these patients is obviously long so our colleagues have rightly asked themselves the question of how nociceptive ionizing radiation is for CT or for the infusion of radioactive drugs, performed for followups and to what extent these treatments can , develop other types of neoplasms. In fact, several articles on this topic have been published in the literature, cited by colleagues, and the conclusion we can draw from them is that a greater risk of developing a secondary neoplasm due to exposure to ionizing radiation for tests conducted every three months is true. It is not certain that this is the case for tests carried out over longer periods. The researchers stuck to the guidelines, however they did not mention the dosage of chromogrania A, enolase, which are nevertheless useful. MRI can be useful for limiting radiation exposure, as also mentioned in the paper, and therefore certainly needs to be developed. the conclusions are consistent with the subject of discussion as well as the limitations they have extensively discussed. The bibliography is consistent and correct, the English can be made more fluent, tables and figures are consistent.

Thank you. We agree that the latency period between ionizing radiation exposure is an important limitation that was not accounted for in our methodology. We have added a line about this to our limitations paragraph. We also feel that increasing the latency period between doses of ionizing radiation exposure is a helpful and practical suggestion for clinicians that could further reduce risk of secondary malignancy and therefore we have added this to our discussion (lines 247 to 249)

We agree use of biochemistry is an important component of surveillance in GEP NETs. Society-derived guidelines also vary in their recommendations for how biochemistry should be utilized, and the burden associated with these investigations (with respect to patients and health care resources) certainly deserves consideration. However,  assessing practices around use of biochemistry was beyond the scope of this project (particularly as our center

most closely adheres to the CommNETs guidelines which recommend against routine biochemistry in this setting and therefore we expect practices around this to be highly variable). We have added at statement to lines 47-48 to clarify that recommendations for biochemistry are included in some surveillance protocols.